Effect of nest age and habitat variables on nest survival in Marsh Harrier (Circus aeruginosus) in a fishpond habitat

Zaremba Urszula uz525@stud.uph.edu.pl
Kasprzykowski Zbigniew
Golawski Artur
Faculty of Exact and Natural Sciences, Siedlce University of Natural Sciences and Humanities , Siedlce , Poland
Harrison Xavier
Electronic publication date: 2020 Sep 9
Publication date: 2020
Volume: 8
Electronic Location ID: e9929
Received 2020 Mar 30; Accepted 2020 Aug 21
Copyright: © 2020 Zaremba et al.
Copyright year: 2020
Copyright holder: Zaremba et al.
License: This is an open access article distributed under the terms of the Creative Commons Attribution License, which permits unrestricted use, distribution, reproduction and adaptation in any medium and for any purpose provided that it is properly attributed. For attribution, the original author(s), title, publication source (PeerJ) and either DOI or URL of the article must be cited.
License URL: https://creativecommons.org/licenses/by/4.0/

Keywords: Breeding time, Fishpond, Nest site selection, Predation risk, Daily survival rate

Funding: Siedlce University of Natural Sciences and Humanities The study has been supported by Siedlce University of Natural Sciences and Humanities. The funders had no role in study design, data collection and analysis, decision to publish, or preparation of the manuscript.

==============================
Background

One important anti‐predator strategy adopted by birds involves nest site selection and timing of breeding. Nest-site selection by marsh-nesting birds often involves nest concealment and water depth as key features influencing nest survival. Marsh Harrier (Circus aeruginosus) is an obligate ground nester, which sets it apart from other raptors. The aim of the present study was to identify for the first time possible temporal and habitat factors affecting nest survival in Marsh Harrier. Understanding features which affect nest survival are essential for assessing relevant conservation strategies.

Methods

To understand the relative contributions of different temporal and habitat variables to brood losses, it is useful to determine the daily survival rate (DSR). We examined 82 Marsh Harrier nests located on fishponds in eastern Poland, where predation is the main cause of nest loss. Six habitat variables were measured for each active nest. DSR was calculated using known-fate models with the RMark package.

Results

The best-supported model predicted that DSR decreased with nest age and was positively affected by the water depth and the diameter of reed stems, but not by the height or density of vegetation at the nest site. The distances of nests to the fishpond dyke and to open water received no support in the models. The chances of nest survival were lower if a neighbouring nest had been depredated. This result suggests that the Marsh Harrier is more susceptible to mammalian than avian predation and confirms the high level of predator pressure in fishpond habitats.

Introduction

Nesting habits of Harriers (Circus spp) sets them apart behaviourally from other raptors—they are the only diurnal birds of prey which nest on the ground. Species across this genus share more unique characteristics which makes them highly adapted to life style in open grasslands and wetlands such as ability to transfer prey in aerial food-pass or foraging at high resolution acoustical abilities (Simmons & Simmons, 2000). Generally, ground nesting among diurnal birds of prey is sometimes observed in areas where predation pressure is relaxed for instance in isolated areas or areas without human disturbance (Ellis et al., 2009). The Marsh Harrier (Circus aeruginosus) is a migrant species, which prefers semi-open habitats with extensive reedbeds (Alves et al., 2014). The typical biotopes of Marsh Harriers are the shores of lakes and fishponds covered with reed, peat bog areas covered with reed and willow as well as small midfield ponds in agricultural-wetland complex mosaic (Witkowski, 1989). Marsh Harrier build nests in aquatic vegetation above the water surface and exceptionally also in cereal fields or meadows. This unique nesting behaviour makes this species a particularly interesting object of study in the context of factors affecting the risk of nest loss, especially those resulting from predation. The influence of habitat features on predation risk is an aspect still largely unexplored in this species.

Predation is one of the most important selective pressures in nature, and birds display a variety of behavioural traits that appear to be adaptations to prevent predators from detecting their nests (Caro, 2005; Ibáñez-Álamo et al., 2015; Pestana, Mateus-Barros & Guillermo-Ferreira, 2020). Anti-predator strategies adopted by birds involve direct effects of parental behaviour (nest defence) as well as indirect ones, such as the decision where (nest site selection) and when (timing) to breed (Lima, 2009). Numerous studies have shown that, for many typical marsh-nesting birds, water depth and emergent vegetation are key features influencing nest site selection and nest survival (Sutherland & Maher, 1987; Polak, 2007; Austin & Buhl, 2011). Nesting success tends to increase with increasing depth of water under the nest (Albrecht et al., 2006; Polak, 2016). It has been demonstrated that water acts as a barrier and limits access to a nest for many terrestrial predators (Jobin & Picman, 1997; Hoover, 2006). Vegetation cover is expected to reduce transmission of auditory and visual cues from a nest to potential avian and mammalian predators, so the better a nest is concealed among reedbed vegetation, the lower the risk of its being detected. Vegetation features such as height (Jedlikowski, Brzeziński & Chibowski, 2015; Batáry & Báldi, 2005) and density (Kristiansen, 1998; Polak, 2016) contribute to nest concealment and positively influence nest survival in marsh nesting birds. Moreover, some studies have shown that factors associated with macro-scale nest placement, such as the distances from the nest to open water and the land-water margin, also affect the risk of nest loss (Brzeziński, Jedlikowski & Żmihorski, 2013; Jedlikowski, Brzeziński & Chibowski, 2015). Besides nest concealment, another factor affecting nest survival could be the fate of the nearest neighbouring nest. If a predator has found one nest in an area, there is a greater chance that it will locate similar nests close to one another (Kristiansen, 1998; Kasprzykowski & Polak, 2014). In order to better understand the relative contributions of different temporal and nest site characteristics to nest survival, it is useful to determine the daily survival rate (DSR), that is the probability that a nest survives a single day (Dinsmore, White & Knopf, 2002). To date, daily nest survival models have been used mainly for waterfowl, shorebirds, and passerines that nest on or near the ground (Davison & Bollinger, 2000; Grant et al., 2005; Smith & Wilson, 2010); only a few studies of raptors have analysed this pattern (Brown & Collopy, 2008; Brown et al., 2013; Crandall, Bedrosian & Craighead, 2015; Segura & Bó, 2018). The aim of the present study was to investigate for the first-time key factors affecting the nest survival rate in Marsh Harrier in wetland habitats with varied vegetation structures and hydrological regimes. Vegetation and wetland characteristics are important factors affecting nest site selection in this species (Stanevičius, 2004; Němečková, Mrlík & Drozd, 2008). In addition, nest site selection appears to be affected not only by nesting habitat quality but also by the presence of conspecifics using the nearby wetland as a foraging area (Cardador, Carrete & Manosa, 2011).

In the present research, answers were sought to the following questions: (1) Does nest age influence Marsh Harrier DSR in a wetland habitat? (2) Does the Marsh Harrier nest DSR pattern differ from that of birds well adapted to nesting in aquatic environments? (3) What is the effect of different habitat characteristics on Marsh Harrier nest DSR? (4) Is the fate of the nearest neighbouring Marsh Harrier nest associated with nest DSR? To address these questions, we assessed the relevance of nest age, the fate of neighbouring nests and habitat variables at nest sites as possible predictors of the daily nest survival rate. Thus, we anticipated a higher DSR for better concealed nests over deeper water and situated in dense vegetation. We also hypothesised that Marsh Harrier DSR would decrease with increasing nest age and depend on neighbouring nest success. Determining these patterns of nest survival may improve our understanding of predator-prey interactions. It is also important for understanding raptor population dynamics, as reliable estimates of nest survival are essential for assessing relevant conservation strategies

Materials and Methods

Study area

The study was conducted over five breeding seasons (2008, 2009, 2011, 2018 and 2019) in eastern Poland on four fishpond complexes: Siedlce, Rudka, Szostek, Mościbrody (52°05′–52°11′N, 21°58′–22°18′E); all are mainly used for the commercial breeding of Common Carp Cyprinus carpio. Pond areas varied from 65 to 203 ha and were surrounded by a mixture of arable fields (50–60%) where mostly cereals were cultivated, meadows (25–30%) and forested areas (5–25%). All four fishpond complexes were located within 20 km of each other. Most of the ponds were partially covered by tall marsh vegetation consisting of Bulrush (Common Reedmace) Typha latifolia, Common Reed Phragmites australis and Sedges Carex spp. These plants tend to proliferate rapidly, quickly covering the surface of a pond or wetland, thus creating a suitable breeding habitat for Marsh Harrier. The ponds were similar in water depth but water levels in the emergent vegetation varied from 7 to 92 cm in spring, falling as the breeding season progressed.

During the fieldwork, the presence of several possible opportunistic predators of aquatic birds’ nests were recorded: the invasive American Mink Neovison vison and Raccoon dog Nyctereutes procyonoides, and the native Wild Boar Sus scrofa, Red Fox Vulpes vulpes, European Otter Lutra lutra, European Badger Meles meles, Magpie Pica pica, Raven Corvus corax and White-tailed Eagle Haliaeetus albicilla. With its rapid colonisation of Poland in recent years, American mink is currently considered the biggest threat to the native waterbird fauna (Brzeziński et al., 2019a, 2019b).

Field procedures

At the beginning of each breeding season, each study pond was visited at 1–3-day intervals between mid-April and mid-May to locate breeding pairs and nests. The observations were made with 8 × 42 binoculars from the fishpond dyke. The birds were observed carrying nest material to the emergent vegetation belt and during aerial food-passes near their potential nest site. After selecting a potentially favourable site, the observers inspected the vegetation belt on foot along fixed line transects. When located, the nests were numbered and their positions recorded with a hand-held GPS unit. A total of 82 nests were discovered in the study areas—27 during the egg-laying phase, 53 during egg incubation and 2 during the early nestling period. To minimise disturbance, each nest was visited at 5–7 day intervals to determine clutch size, hatching date, nest fate and the number of live chicks. The first-egg laying date was calculated on the assumption that eggs are laid at 2-day intervals, and that incubation starts after the laying of the first egg and lasts for an average of 33 days (Witkowski, 1989). A nest was considered to have been depredated when clear signs (egg shells or nestling feathers) were found or the nest was found empty before the predicted date of fledging. A successful nest was defined as one where at least one young bird survived up to 35 days old (Witkowski, 1989). To be sure of this success, additional inspections of such a nest were made 5 days before and 3 days after this period. Six habitat variables were obtained for each active nest along with the fate of the nearest neighbour nest (Table 1). All the vegetation measurements (height, diameter and density) were made in 100 × 100 cm quadrats placed around the nest during the first visit. The distances of a nest to the fishpond dyke and to open water was measured using GPS equipment. All operations were conducted as part of the ‘Research of birds in the disclimax ecosystems’ programme, approved by the Institute of Biological Sciences, Siedlce University of Natural Sciences and Humanities (number of approval: IB.5030.8.2018). The study took place in compliance with current Polish Law and was approved by Ministry of the Environment (permit number: 425/2019) and also by the Regional Directorate for Environmental Protection in Warsaw (permit number: WSTS.6401.34.2018.MO).

Table 1 Variables obtained at ponds with active Marsh Harrier nests.

Code	Meaning	
Water	Estimated water depth (cm) at the centre of the plot with 1-cm precision	
Height	Mean height of five flowered dry reed stems chosen randomly with 10-cm precision	
Diameter	Mean diameter of 10 reed stems chosen randomly with a calliper precision of 0.1 mm	
Density	Number of stems within a 1 × 1 m square	
Distdyke	Distance (m) to the fishpond dyke	
Distow	Distance (m) to open water pool	
NNFate	Fate of the nearest neighbour nest: success (1) and loss (0)	

Statistical analyses

Daily survival rate, the probability that a nest will survive a single day, was calculated using known-fate models with the RMark package (Laake, 2019). RMark is an R package (R Core Team, 2019) that provides a formula-based interface for the MARK programme (White & Burnham, 1999). The analysis covered only nests which succeeded or were depredated (N = 82). The dates were scaled such that day 1 was the day when the first nest was found and day 84 was the day the last nest was checked. Hence, the 84-day nesting season was defined as beginning on 30th April and ending on 22nd July. The season thus consisted of 83 intervals, which represent an 83-day nesting cycle with each interval equivalent to 1 day. We therefore modelled DSR as a function of temporal (nest age), the fate of the nearest neighbour nest and habitat variables (water depth under the nest, density of reed stems, height and diameter of vegetation, distance to open water and distance to the dyke). The multi-co-linearity of habitat variables was checked—the correlations between them were <0.4. We constructed models of nest survival that incorporated combinations of individual covariates, and compared them to the null model of constant survival rate S(.). The set of competing models was based on a combination of factors assumed a priori to affect DSR.

The better concealed nests were expected to have a greater chance of survival. We used an information-theoretic approach (AIC) to compare the competing models (Burnham & Anderson, 2002) and analysed model support using the Akaike’s information criterion with small-sample bias adjustment (AICc) value, which corrects for small sample sizes and evaluates the strength of evidence for each model using normalised weights (wi). We applied selection approach suggested by Richards (2008), in which more complex variants of any model with a lower AICc value are excluded from the candidate set. The models selected with the smallest AICc as being the best of all the models compared, where the models were within a ∆i AIC of 2.00, were considered to be equally supported (Burnham & Anderson, 2002).

Results

We monitored a total of 82 Marsh Harrier nests: 52 were successful and 30 were depredated. The breeding success (at least one fledgling produced) over all the study years was 64%; success was the highest (84%) in 2011 and the lowest (9%) in 2018, when only one pair successfully raised young (Table 2). Eight (27%) of the nests were predated during the egg-laying stage, 16 (53%) during incubation and 6 (20%) during the nestling period. The average height of the reed stems in the nesting squares was 1.9 m (SD = 39.2; N = 82; range 106–300) and the average diameter of the shoots was 7.3 mm (SD = 1.8; N = 82; range 3.3–12.1). The density of stems varied between 31 and 191 (mean = 86.9; SD = 31.1; N = 82). The level of water at the nest at the beginning of the nesting season varied between 21 and 92 cm (mean = 50.4; SD = 15.6 cm; N = 82). The average distance of a nest to the fishpond dyke was 69.8 m (SD = 49.1; N = 82; range 15–233) and the average distance of a nest to open water was 51.0 m (SD = 50.2; N = 82; range 1–278).

Table 2 Sample size of Marsh Harrier nesting attempts by year.

Year	2008	2009	2011	2018	2019	
Depredated nests	6	5	3	11	5	
Successful nests	12	16	15	1	8	
Total	18	21	18	12	13	

In the null model, DSR calculated for all nests was 0.992 ± 0.001 SE (95% CI [0.989–0.994]). The analysis revealed that both temporal and habitat variables as well as fate of neighbouring nest affected Marsh Harrier DSR (Table 3). The top model of the 14 a priori models with the highest ranking (AICc = 194.1) received 80% support (sum of wi, Table 3) and included combinations of nest age, fate of neighbouring nest and two habitat variables (diameter of reed stems and water depth). The best-fitted model with the lowest AICc predicted that Marsh Harrier DSR decreased with nest age (Fig. 1). The habitat factors with the greatest influence on the likelihood of nest depredation were the diameter of reed stems around the nest and water depth: in both cases, DSR gradually increased with these parameters (Fig. 1). In addition, the chance of nest survival was smaller if a neighbouring nest was depredated (Fig. 2). The analysis also showed that factors such as density of vegetation, distance to open water and distance to the fishpond dyke had no effect on the survival of Marsh Harrier broods. There was greater support for the null model of constant survival rate S (.).

Table 3 Summary of candidate DSR models of Marsh Harrier in eastern Poland.

The number of estimated parameters (npar), Akaike’s information criterion with small-sample bias adjustment (AICc), delta (∆i) representing the difference in AICc between the current and the most appropriate model, and the model weight wi are shown.

Candidate model	npar	Deviance	AICc	(∆i) AICc	wi	
S(~NestAge+Diameter+Water+NNFate)	5	184.035	194.1	0.00	0.795	
S(~NestAge+Diameter+Water)	4	189.179	197.2	3.14	0.165	
S(~NestAge+Water+Distow)	4	193.011	201.0	6.97	0.024	
S(~NestAge+Water)	3	196.880	202.9	8.83	0.009	
S(~NestAge)	2	200.342	204.3	10.3	0.004	
S(~Diameter+Water)	3	202.913	208.9	14.9	0.000	
S(~Diameter)	2	205.989	209.9	15.9	0.000	
S(~NNFate)	2	206.401	210.4	16.4	0.000	
S(~Water)	2	206.622	210.6	16.6	0.000	
S(~Height)	2	208.680	212.7	18.6	0.000	
S(~.)	1	212.351	214.4	20.3	0.000	
S(~Distdyke)	2	211.448	215.5	21.4	0.000	
S(~Distow)	2	211.605	215.6	21.5	0.000	
S(~Density)	2	212.047	216.1	21.9	0.000	

Figure 1 Model-averaged estimates of daily nest survival for Marsh Harrier in eastern Poland showing the effect of nest age (A), mean diameter of vegetation (B) and water depth (C).

The solid line represents the daily survival rate estimated using beta parameters from the bestfit model. The grey areas around the line represent the 95% confidence intervals for the estimated daily survival rate.

Figure 2 Median probability of nest survival (points) with 95% Cl (whiskers) for nests for which the nearest neighbouring nest was successful (success) or depredated (loss).

Discussion

Nest age

Our results showed that Marsh Harrier DSR was not constant from the egg phase to fledging, decreasing gradually with nest advancement. It is likely that multiple factors influence age-specific patterns of nest survival. One potential explanation of this pattern is that after hatching, parents and young provide more behavioural cues at the nest, which increase the possibility of its being detected by predators. Increased parental visits to the nest may raise the risk of nest predation (Martin, Scott & Menge, 2000). Indeed, in the late nesting period, Marsh Harrier parents make more food deliveries daily (Kitowski, 2006). However, as nest age increases, adults invest more in the nest and typically intensify defensive behaviour (Caro, 2005); this pattern has also been confirmed in other birds of prey (Carrillo & Aparicio, 2001; Sergio & Bogliani, 2001). Despite the fact that Marsh Harrier actively defend its nests (Kitowski, 2006), this does not compensate for increased predation. One possible explanation is that with nest advancement, parents need to make more frequent foraging flights to provide for their offspring and spend less their time defending the nest. The times when the parents are absent are when the nest is more vulnerable to predation. Decreasing DSR may also reflect a cumulative risk: the longer a nest is active, the more likely it will lose eggs to predation. In Marsh Harrier, the period from the start of incubation to the fledging of the young birds is relatively long in comparison with other species nesting in the same fishpond habitat. For example chicks of Eurasian Bittern (Botaurus stellaris) leave the nest at the age of just 2 weeks post hatching, that is before reaching full independence (Kasprzykowski & Polak, 2012). The female continues to care for the young, which hide in vegetation near the nest until fully fledged. This could be an adaptive strategy diluting the risk of detection by predators and preventing DSR from decreasing with nest age in this species. In contrast, Marsh Harrier chicks cannot leave the nest until they are capable of flight: this species is therefore especially vulnerable to detection by predators with increasing nest advancement.

Nest concealment

In our study, the diameter of reed stems but not the density or height of vegetation influenced Marsh Harrier nest DSR. Nests built over deeper water, where reed stems were thicker, were less vulnerable to predation. Nests located in sites with thicker vegetation may be better protected from predators because the habitat structure at these sites may improve visual concealment of nest thus reducing probability of nest detection. Vegetation thickness can also improve protection from wind thus reducing olfactory cues to potential mammal predators (Guyn & Clark, 1997). The height of the vegetation was not a variable significantly improving the chances of brood success in Marsh Harrier; this factor has been proven significant, though mostly for populations of birds depredated by avian predators (Jedlikowski, Brzeziński & Chibowski, 2015). The distances from nests to open water or the fishpond dyke were not significant. This finding is consistent with previous studies on this species (Stanevičius, 2004). It is well-known that ground-nesting birds, including Marsh Harrier, are particularly vulnerable to predation. Thus, obligate ground-nesters have evolved a method of placing their nests in well-concealed, evenly-spaced sites to reduce the likelihood of detection (Redmond, Keppie & Herzog, 1982). We expected the density of vegetation to have an impact on nest DSR of Marsh Harrier, as was found to be the case in other marsh nesting birds (Kristiansen, 1998; Polak, 2016). The reason for the lack of such a relationship is that parental behaviour (nest defence) may compensate for any effects of insufficient nest cover (Lima & Dill, 1990). Marsh Harrier is considered a top avian predator of wetland habitats and actively defends its own nests with alarm calls and physical attacks (Witkowski, 1989).

Water depth

A strong positive relationship between water depth and nest survival has been observed in marsh-nesting birds, for example in Eurasian Bittern (Polak, 2016) and Common Pochard Aythya ferina (Albrecht et al., 2006). In the present study, Marsh Harrier nests located at sites with deeper water exhibited the same trait. Such nests were particularly successful, because water presents a barrier to many mammalian predators (Koons & Rotella, 2003). It is worth noting that the water level is not constant throughout the breeding season. Previous studies have shown that predation rates for wetland nesters decreased with increasing water depth (Purger & Mészáros, 2006). On the other hand, water depth was not important for the DSR of either Little Crake Porzana parva or Water Rail Rallus aquaticus, as their main predators are mostly avian (Jedlikowski, Brzeziński & Chibowski, 2015). This pattern suggests that Marsh Harrier builds its nests over deeper water because they are more susceptible to predation by mammals than by birds. The relationship between a low water level and mammalian predation was demonstrated in the Netherlands, where Red Fox was a frequent predator of Marsh Harrier nests in dry reedbeds (Dijkstra & Zijlstra, 1997). In his study of a breeding population of Marsh Harrier in the Barycz Valley, Witkowski (1989) also noted that dried out ponds were avoided, although the same ponds containing water could have one of the highest densities of nesting pairs. This may suggest that locating nests at deep water sites is an important antipredator strategy in this species. During studies in eastern Poland 30 years ago, Buczek & Keller (1994) highlighted corvids and mustelids as being the main predators of Marsh Harrier nests on retention reservoirs (mostly resembling neglected fishponds) and bogs, respectively. This was further explained by the low water level in bogs, decreasing during the course of the season, which enabled mammalian predators to penetrate reedbeds. Since that study, predator and prey interactions could well have changed significantly, following the spread of non-native invasive predators such as American Mink. This has been confirmed by recent research, which links the declines of several waterbirds and semi-aquatic mammals with the colonisation of Poland by American Mink (Brzeziński et al., 2019b). Even though water is not a barrier for American Mink, the occurrence of this invasive species may also be having an impact on Marsh Harrier, making it even more vulnerable to mammalian predation. But to explain this possible shift, further studies will be needed to evaluate the causes of nest loss in Marsh Harrier in greater detail.

Neighbour fate

The influence of nearest neighbour fate on nest success has been addressed in observational and experimental studies, but their results are inconsistent (Ackerman, Blackmer & Eadie, 2004; Ringelman, Eadie & Ackerman, 2012). In our study, the chances of nest survival were lower if a neighbouring nest was depredated. A similar pattern is observed in other marsh nesting birds like Common Pochard Aythia ferina (Folliot et al., 2017). In other wetland bird species, nearest-neighbour nests tend to share the same fate but only when they form clusters, which survive or are depredated as a group (Ringelman, Eadie & Ackerman, 2012; Kasprzykowski & Polak, 2014). The reason why neighbours’ fates influence a nest’s fate is that predators use area-restricted searching and employ search images to find similar nests nearby. However, other studies on waterfowl have not detected any influence of nearest neighbour fate on the success of artificial or natural nests (Ackerman, Blackmer & Eadie, 2004). The reason for this difference could be the predator community and the spatial scale at which potential predator species operate that is large vs. small home ranges (Schmidt & Whelan, 1999).

Conclusions

Our study has shown that the DSR of Marsh Harrier is influenced by both temporal and certain habitat variables. DSR is the highest at the beginning of the nesting season and decreases gradually with nest advancement. Water depth and the mean diameter of vegetation at the nest site were the habitat variables influencing Marsh Harrier DSR. This result suggests that Marsh Harriers are more susceptible to mammalian than avian predation. The relationship between nest survival and nearest neighbour nest fate seems to confirm the high level of predator pressure in fishpond habitats. Further studies are needed in order to gain a better understanding of the accessibility of wetland birds’ nests to terrestrial predators in the context of biological invasions.

Supplemental Information

Supplemental Information 1 Raw data containing information on Marsh Harrier nests.

Includes temporal and habitat variables.

Click here for additional data file.

Supplemental Information 2 R code for statistical analysis.

Click here for additional data file.

We are grateful to Karol Bosek, Rafał Kuropieska, Daniel Paczóski, Marta Szaniawska, Kamil Kryński, Tomasz Pietrzak, Paweł Radzikowski, Monika Budzyńska and Cezary Sadowy for their help in carrying out the fieldwork. We would like to also thank Jay Rotella for helping with the statistical analyses.

Additional Information and Declarations

Competing Interests

Author Contributions

Animal Ethics

Data Availability

The authors declare that they have no competing interests.

Urszula Zaremba performed the experiments, analysed the data, prepared figures and/or tables, authored or reviewed drafts of the paper, and approved the final draft.

Zbigniew Kasprzykowski conceived and designed the experiments, performed the experiments, analysed the data, authored or reviewed drafts of the paper, and approved the final draft.

Artur Golawski performed the experiments, authored or reviewed drafts of the paper, and approved the final draft.

The following information was supplied relating to ethical approvals (i.e. approving body and any reference numbers):

All operations were conducted under the programme ‘Research of birds in the disclimax ecosystems’ approved by the Institute of Biological Sciences, Siedlce University of Natural Sciences and Humanities (number of approval: IB.5030.8.2018). The study fulfilled the current Polish Law and was permitted by Ministry of the Environment (approval number: 425/2019) and Regional Directorate for Environmental Protection in Warsaw allowed for this research project by the letter (number of approval: WSTS.6401.34.2018.MO).

The following information was supplied regarding data availability:

Raw and code are available in the Supplemental Files.

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
