# Peer review of "Effect of nest age and habitat variables on nest survival in Marsh Harrier (Circus aeruginosus) in a fishpond habitat"

_PeerJ, doi:10.7717/peerj.9929_

## Round 0.1 · original submission · Major Revisions

· Academic Editor

Major Revisions

Your manuscript has now been assessed by two expert reviewers. We all agree that this study is interesting and has the potential to make a valuable contribution to the literature. However, we also all have identified a number of concerns that need to be addressed in a revision. Reviewer 1 has annotated the manuscript to helpfully provide numerous areas for improved framing of the question, clarity of writing, and citations to support your arguments. Reviewer 2 and I have some analytical queries and suggestions for improvement, and I detail mine below:

1) You talk in the introduction about the importance of separating the potential confounds of nest age and day of season, but this seems odd given that it’s not clear your study has managed this. To address this issue, I would expect to see nest age and nest initiation date in the same model - but centred to a baseline anchor to ensure the two variables aren’t correlated. This may be what you’ve done, but the methods aren’t very clear.

2) I agree with reviewer 2 that the analyses seem a bit too simplistic given the data collected. I would expect a signal of spatial autocorrelaton in your data where there might be ‘hotspots’ of failure and success, and so success of a focal nest may be dependent on that of nests near to it. This may occur, for example, because a predator finding one nest will be more likely to find one nearby than one the other side of the study site.

3) With regard to your statistical analysis, I didn’t see any tests for collinearity among your explanatory variables, and I would expect traits like water depth, reed height, and reed diameter to all be correlated. This would suggest they shouldn’t be in the same model(s)

4) In the discussion and conclusion, you talk about both time and nesting age affecting DSR, but these were in separate, competing models. The fact that both models had similar AIC suggests the two terms are correlated and fighting for the same variance. So I don’t think there is evidence in your study that both are important.

5) Four separate figures for the prediction curves seems excessive. The figures would in fact be much clearer if combined into one four panel figure with panel headings of what the predictor variable was. And please don’t scale the y axes to make the effects appear more meaningful. Figures 1 and 4 are particularly guilty of this. Reviewer 1 has also identified this issue.

6) You conducted the study over multiple years but have seemingly lumped all of the data together to conduct the analysis, even though you acknowledge that annual fledging success was highly variable. This seems very odd. I’d also like to see a table of sample sizes of nest-per-year to help contextualise the analysis and power of your study.

·

Basic reporting

The manuscript reports the daily survival rates of Marsh Harrier (Circus aeruginosus) nests in relation to different environmental and temporal variables. It is a novel and well written contribution, although I find some problems in the general structure that concern me. I make numerous comments throughout the ms, among the most important ones I highlight that in the Introduction section, the background is not well ordered and does not allow asking clear questions about the objectives of the study. Also in Discussion I find some problems that deserve to be addressed.
Since English is not my native language, I have made only a few comments regarding writing.
The literature cited also worries me, since it is too broad and linked to birds with very different life stories. In this sense, I suggest adjusting Introduction and Discussion to raptors, preferably those that nest in the ground.

Experimental design

As I mentioned above, the background does not clearly link to the question in this study. Methods are well developed, I only make a few small observations. The results are clearly presented, I only make suggestions about the figures.

Validity of the findings

Discussion and Conclusions need to be revised (see comments in the text).

Additional comments

I believe the manuscript is potentially publishable after major changes in the Introduction and Discussion sections. Most of the comments are represented in the previous items.

Reviewer 2 ·

Basic reporting

Zaremba et al. analyses potential factors affecting daily survival rates of marsh harriers at nest in relation to predation. They focus in three types of factors: microhabitat characteristics of the nesting place, time of breeding and nest age. For this, they surveyed a total of 82 marsh harrier nests in four fish ponds in Poland in years 2008, 2009, 2011, 2018 and 2019. The paper is clear and well written and the scope is valuable, as information on nesting predation and its main drivers has been largely unexplored for the marsh harrier. However, I have some concerns about the data and statistical analyses.

Experimental design

My main concern is related to potential effects of non-independence of the data used for analyses. Non-independence might be occurring at different levels in this study (repeated years, repeated breeding locations (and maybe breeders in those locations) among years, repeated sampling in the four fishpond complexes considered and non-independence related to spatial autocorrelation of neighbouring nests). This is important as, if for example, predation is mainly concentrated in a few locations (with similar habitat characteristics given their non-independence), you might find a significant relationship with such characteristics even if they are not the ultimate reason of differences in DSR. To deal with this, authors should account for non-independence better. There are different ways to account for this: e.g., include breeding sites, years or locations as variables (random terms if possible) in models, include the X and Y coordinates, and all of their interactions terms up to the third order (Legendre & Legendre, 1998) in models, or apply Moran I test statistics to analyses non-independence in model residuals. Authors could also provide a better description of the distribution of nests depredated across fishponds, years or locations.

Second, the authors find an effect of time and nest age on DSR. However, detection probability of nests seems to be in turn affected by time. As authors stated, “a total of 82 nests were discovered in the study areas – 27 during the egg-laying phase, 53 during egg incubation and 2 during the early nestling period”. This suggests that higher DSR early in time, could be simply due to the fact that detection is lower early in the season, and as so nests depredated (i.e., early predations are not computed, because nests are simply not detected). To separate the effect of time on DSR from sampling biases, authors should account for detection differences across time in models.

Third, I have also concerns related to the variables used in analyses. Previous evidence (e.g., Cardador et al. 2011 Anim. Consev.) has shown that nesting site selection of marsh harriers is influenced by habitat characteristics at two scales (nesting site and foraging habitat). Both of these scales are, in turn, expected to affect predation risk at nests. Particularly, foraging habitat quality is expected to have a direct influence on time dedicated to parental care. Thus, considering broader scale variables might largely improve analyses.

Fourth, and related with the comment above, according to deviance information provided in table 2, authors’ final models do not fit very well the data (only 11% of deviance explained). This suggests that authors might be missing some relevant factors explaining DSR. Habitat at broader scale can be one of this. But other factors such as conspecific densities or individual quality could be also relevant. Authors should consider including some of such variables to improve model performance. At the very least, the low fit of models and its potential reasons should be better discussed.

More specific comments:

Line 99: Could you provide some more information on fishpond complexes? For e.g., potential differences in their surrounding habitat or in density of predators,…

Line 125-126. “A nest was considered to have been depredated when it was found empty before the predicted date of fledging, which is 35 days of age according to Witkowski (1989)”. What was the previous contact - 7 days earlier or more? This is important because, particularly in late stages, you might also be confounding predation with fledging. Apart from the nest empty, did you detect any other sign of predation? Also, for depredated nests, when was the late contact with breeders? Is it possible that for some predations, parents abandoned first and then the nest is depredated?

Lines 223-224: “The reason for this pattern could be that the earliest breeding birds are often older and more experienced than late nesters”. In Cardador et al. 2012 Plos one, we found evidence for this. Related to this, did you find differences in the phenotype of birds breeding earlier (i.e., grey vs. brown males)?

Figures: Providing raw data apart from model predictions might be more informative.

Validity of the findings

Results of the study are well presented and discussed, however they might be flawed by some methodological issues commented above.
I hope my comments to improve it could be of help.

---

## Round 0.2 · Minor Revisions

· Academic Editor

Minor Revisions

Your manuscript has now been reassessed by one of the original reviewers. The manuscript is certainly improved from the original submission, but a couple of outstanding issues remain.


- You talk about nest predation being the most common cause of breeding failure in birds, but this feels like a gross generalisation, especially as there are many species whose productive attempts fail (e.g. cooperative breeders in unpredictable environments). 30 year old references should be supported by more recent evidence too.

In general, I have an issue with talking about nest predation as 'breeding failure'. Breeding failure can mean failure to lay eggs at all, eggs being unviable, fledglings not surviving due to bad weather or poor food availability etc. I think it's important to be able to distinguish between these terms.

- There seems to be a real push to frame the introduction as predation being a huge selective pressure for all birds to justify the current study. But I would argue that it would be better to frame it in terms of understanding population-limiting processes in this particular raptor. Then it doesn't matter if predation isn't at all important in other birds (for the sake of argument). It's about what's relevant to this species. The introduction needs to be reframed and streamlined, which will have the advantage of no longer requiring broad sweeping statements about where predation ranks as a threat to ALL birds

L29: It's more common to talk about the 'best-supported model'

L31: 'not important factors' feels too vague. I think you mean received no support in the models

Table 3: Your second-best-supported model is a more complex version of a model with better AIC support, and so should be removed under the nesting rule (see Richards 2008 https://besjournals.onlinelibrary.wiley.com/doi/abs/10.1111/j.1365-2664.2007.01377.x)


Reviewer 1:
Discussion:
First paragraph: Certainly, increasing the number of visits to the nest could attract more predators, but it is not clear to me the authors' association with nest defense in these later stages of the nest cycle, it is somewhat contradictory. Explain more clearly please.
Second paragraph: Regarding the effect of nest age, the sentence ‘This may suggest that Marsh Harrier is not so well adapted to nesting in wetland environments as other species doing so’ seems somewhat vague to me.
Nest concealment: too much emphasis on negative results and too little discussion of positives.

---

## Round 0.3 · Minor Revisions

· Academic Editor

Minor Revisions

Many thanks for making the required changes to the manuscript. The framing of the introduction is now much more congruent with the data in the paper. There are only a few minor issues to clear up, after which I'd be happy to recommend your paper for publication

Line numbers refer to your track changes version:

L48: "The" Marsh Harrier
L50: Marsh Harrier[s]
L52: Marsh Harriers build
L55: "The" influence
L241: "The" top model
Line 242: What is delta-I? It reads as if you are not considering a top model set any more i.e. delta6? On L224 you still talk about a delta 2 model set, but have removed this information from the results. Some clarification is required here.
L370: Marsh Harriers are

---

## Round 0.4 · accepted · Accept

· Academic Editor

Accept

I'm now happy to recommend your manuscript be accepted for publication.